# Effect of Vanadium Layer on Microstructure and Properties of TC4 (Ti-6Al-4V)/TiAl (Ti-48Al-2Cr-2Nb) Dissimilar Metals Produced by Laser Additive Manufacturing

**Haijiang Wang, Zhanqi Liu \*, Jianhui Liang, Wei Wei and Guili Yin**

School of Materials Science and Engineering, Liaoning University of Technology, Jinzhou 121001, China; ww1510545@163.com (H.W.); liangjianhui2000@126.com (J.L.)
\* Correspondence: clliuzhanqi@lnut.edu.cn

**Abstract:** Dissimilar metal samples of TC4/TiAl were successfully prepared by laser additive manufacturing (LAM) technology, with pure vanadium as the interlayer. The microstructure, phase composition, element distribution and mechanical properties at the interface of TC4/V and TiAl/V were analyzed by optical microscope (OM), scanning electron microscope (SEM) and backscattering diffraction (EBSD). The experimental results showed that the interface microstructure of TiAl/V is mainly composed of $\gamma$, $\alpha_2$ phase and V solid solution. The microstructure of the TC4/V interface is mainly composed of $\beta$-Ti and V solid solution. There are no holes, metallurgical defects or microcracks at the above two interfaces, and the interface is bonded well. With the increase in the number of deposition layers, the interface bonding depth increases, and its thickness increases from 30 μm to 80 μm. The mechanical properties tests showed that the tensile strength and elongation of dissimilar metals with two layers of V interlayer TC4/TiAl are the highest, and their values are 483 MPa and 0.35%, respectively. Compared with the one-layer V intermediate layer sample (tensile strength 405 MPa, elongation 0.24%), the tensile strength and elongation are increased by 19.2% and 45%, respectively. The tensile strength and elongation of dissimilar metals in three-layer V interlayer TC4/TiAl are the lowest, and their values are 350 MPa and 0.16%.

**Keywords:** laser additive manufacturing; vanadium layer; dissimilar metals; TC4/TiAl; microstructure; mechanical properties

## 1. Introduction

TiAl-based intermetallics are considered one of the most promising candidates to replace nickel-based superalloys because of their excellent physical properties (low density, high specific strength, good oxidation resistance and creep resistance). Therefore, TiAl-based metals are a type of high-temperature structural material with broad prospects below 750 °C and have become the frontier hotspots of aerospace materials [1–6]. Ti-based alloys (such as TC4) play an important role in the aerospace field. High specific strength can not only reduce the fuselage weight but also meet the most stringent strength requirements. Good high- and low-temperature mechanical properties enable it to have stable comprehensive properties in high- or low-temperature environments [7–11]. The connection of TC4/TiAl dissimilar metal materials can give full play to their respective advantages and has a great application prospect in the field of aerospace engines. An aircraft's engine is the heart of the aircraft. In order to reduce the weight of the engine and at the same time to increase the flexibility of the application of the two metals, the two metals can be combined to apply to the engine; TiAl alloy is used in the aircraft engine's low-pressure turbine blades, TC4 alloy is used in the aircraft engine's turbine disk.

Therefore, the dissimilar metal structure connection of Ti-based alloy/TiAl-based alloy has been widely studied by researchers, and how to achieve a good connection between dissimilar metal materials has become a significant problem.

With the development of science and technology, laser additive manufacturing (LAM) provides a new method to promote the production of complex parts efficiently and flexibly. LAM technology uses the principle of discrete stacking to pre-designed three-dimensional CAD models through layered stacking and cumulative manufacturing processing into a solid. The biggest feature of the forming method is using metal powder as the main forming material, with the help of computer-aided design and manufacturing and the use of high-energy laser beams to directly form the complex structure of the three-dimensional solid parts, after rapid melting and solidification of the formed parts. Additive manufacturing technologies can be divided into two categories: directed energy deposition (DED) and powder bed fusion (PBF). DED technology is the process of delivering a feedstock to a spatial location determined by the energy input in conjunction with the desired deposition point. The combination of laser and powder is the most common deposition process. For example, laser metal deposition (LMD), laser engineered net shaping (LENS) and direct metal deposition (DMD) are all such processes. In the LMD process, powder is fed through a nozzle directly into a melt pool that is melted by the laser beam, enabling rapid melting and solidification. By manipulating the spatial position of the melt pool, a complete 3D part can be built point by point, line by line and layer by layer. Powder is transported by gas to force deposition to the desired location, making it possible to add material in non-horizontal planes. The process allows for the construction of gradient materials by adjusting the scanning and powder delivery rates, while separate nozzles and powder silos offer the possibility of in situ alloy mixing and the manufacture of gradient materials. In this way, objects with tough cores and hard wear-resistant surfaces can be manufactured. In addition, the high flexibility of the technology makes it ideally suited for the fabrication of large mechanical or structural components.

LAM technology is more extensive than other AM technologies, and it is also a more mature direction of AM technology development. It can be said that LAM technology is an ideal manufacturing method for the connection of Ti-based alloy/TiAl-based alloy dissimilar metal materials. Dissimilar metal connection can be realized by optimizing parameters and adding alloy elements on the basis of LAM technology [12–16]. Compared with other additive manufacturing technologies, LAM technology has high manufacturing precision (20 μm), the minimum wall thickness can reach 100 μm, component performance can reach the level of forging with the same composition and component densities are nearly 100%. Therefore, LAM technology is more suitable.

Other joining techniques that can be used for these different materials are brazing and diffusion joining. The brazing method allows for reliable joining of TiAl alloys but is prone to introducing other impurity atoms into the base material or generating a large number of brittle intermetallic compounds in the brazed joints [17]. Song et al. investigated the heterogeneous joining of Ti-6Al-4V titanium alloy and AISI316L stainless steel by diffusion joining using Cu/Nb multilayer interlayers. The results showed that the formation of intermetallic compounds was successfully prevented when the joining temperature was 900 °C or below, and a strong TiA/a-bTi/Nb/Cu/SS diffusion-jointed joint was realized by inserting Cu/Nb multilayer interlayers [18]. As another example, Liu et al. [19] successfully prepared defect-free TA15/$Ti_2AlNb$ dissimilar metal samples by LAM technology. It was found that the phase on the side of the lateral $Ti_2AlNb$-based alloy of TA15 alloy changed into α (lath) + β → α (coarse needle) + β → β + $α_2$ → $α_2$ + O + B2. The composition of the transition region showed a stepped platform distribution, with an average tensile strength of 1025 MPa and an elongation of 7.3%. Chen et al. [20] investigated the microstructure and mechanical properties of electron beam-welded TiAl/TC4 dissimilarity joints by changing the heat distribution and found that the Gibbs free energy of the formation of α-Ti3Al at high temperature was considerably lower than that of TiAl, which was favorable for the formation of two phases. However, the TiAl side heat-affected zone also generated a brittle phase. Feng et al. joined $Ti_2AlNb$ alloy to TC4 alloy by vacuum diffusion bonding. The relationship between the bonding parameters and the bonding microstructure and shear strength was investigated. The results showed that the diffusion of Al, Ti, Nb and V

elements at the bonding interface led to the formation of three reactive layers: a B2/β layer and an α2 layer on the TC4 side and an α2 + B2/β layer on the Ti$_2$AlNb side. The shear strength of the joint reached a maximum value of 467 MPa when the Ti$_2$AlNb alloy and the TC4 alloy were bonded at 950 °C and 10 MPa for 30 min [21]. TiAl and TC4 titanium alloys were subjected to electron beam welding experiments by Chen et al. The effect of aluminum content in the weld on the microstructure and mechanical properties of the joint was analyzed. The results showed that the high content of aluminum leads to the formation of a brittle α2 phase, which reduces the ductility and toughness of the joint. The weld was mainly composed of α2-Ti3Al and α-Ti phases, with small amounts of B2 and YAlx phases when the beam was applied to the contact surface. The content of molten TiAl and TC4 in the joint was approximately 1/3 and 2/3, respectively, and the percentage of aluminum was approximately 28 wt.% [22].

Zhao et al. [23] achieved an excellent combination of TiAl alloy and TC4 alloy by adding a CoCuFeNiTiV0.6 interlayer. It was found that, with the increase in bonding temperature and pressure, the thickness of different diffusion layers on the interface of TiAl increases gradually, and the growth activation energy of the B2 diffusion layer was the highest, which was 209.3 kl/mol. The Cu and Ni atoms diffused into the TC4 matrix and eutectoid reaction took place to form a needle-like structure.

In this experiment, from the phase diagram of Ti-V binary alloy, it can be seen that the V element and Ti can form an infinite solid solution, which can effectively inhibit the formation of brittle phases, and at the same time, V has better ductility, so considering V as the V element is reasonable.

Pure vanadium was designed and manufactured as the interlayer for the diffusion bonding of TiAl alloy and TC4 titanium alloy. Three types of V-layer dissimilar metals with different layers (one layer, two layers and three layers) were prepared by LAM technology. The effect of V-layer thickness on the microstructure and mechanical properties of TC4/TiAl dissimilar metal joints was studied. The experimental results can provide a theoretical basis for the preparation of dissimilar metal parts.

## 2. Materials and Methods

The powders used in the laser additive manufacturing were Ti6Al4V (TC4) titanium alloy powder, Ti48Al2Cr2Nb (TiAl) alloy powder and pure V powder. TC4 and TiAl alloy powders were produced by AVIC Maite Powder Metallurgy Technology Co., Ltd. (Beijing, China). The preparation of TiAl and TC4 powder was carried out by an ion-rotating electrode process. The appearance of the powder was spherical and silvery white. Compared with other shapes, the fluidity of spherical powder was better. The particle size of the spherical powder was 53–120 μm. The chemical composition (wt.%) of Ti48Al2Cr2Nb alloy powder is 31.06 Al, 2.49 Cr, 11.35 Nb, 0.06 O, 0.005 N, and the rest is Ti. The chemical composition (wt.%) of TC4 titanium alloy powder is 6.04 Al, 3.82 V, 0.16 O, 0.03 N, and the residue is Ti. The particle size range of V metal powder is 40 um~180 um and its morphology is irregular and non-spherical, and it was provided by Changsha Tianjiu material Co., Ltd. (Changsha, China).

The LAM forming equipment was produced by Nanjing Zhongke Yuchen Laser Technology Co., Ltd. (Nanjing, China), with the LDM8060 system, composed of an LDF-4000 fiber coupled semiconductor laser, water cooler, three-axis CNC worktable and a four-way powder feeding and printing head. In order to prevent the sample from being oxidized during the molding process, the inert gas chamber uses 99.99% Ar protective gas to ensure that the water and oxygen content in the molding chamber is lower than 50 ppm. Before the test, the surface of the TC4 substrate was polished and cleaned with a grinder in order to remove the oxide film and surface defects on the substrate surface, and it was then cleaned with anhydrous ethanol and acetone to remove impurities such as oil. The laser processing parameters are shown in Table 1. Three groups of thin-walled specimens with a thickness of 0.4 mm were prepared on a TC4 titanium alloy substrate. Its appearance is shown in Figure 1.

**Table 1.** Parameters for laser processing.

| Material | Laser Power (W) | Laser Scan Speed (mm/min) | Powder Feed Rate (g/min) | Spot Diameter (mm) |
|---|---|---|---|---|
| TC4 | 1900 | 540 | 5.6 | 3.5 |
| TiAl | 1400 | 420 | 6.2 | 3.5 |
| V | 1900 | 420 | 6.5 | 3.5 |

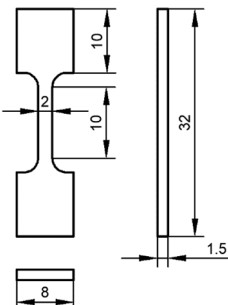

**Figure 1.** Tensile sample diagram.

A sample with a size of 5 mm × 5 mm was cut from the thin-walled sample by WEDM, the surface of the sample was cleaned in acetone solution and then the sample was hot sealed with embedded powder. Terrazzo paper with different particle sizes (600#, 800#, 1000#, 1200#, 1500#, 2000#) was ground in a cross way. The sample was polished to a mirror finish with diamond spray polishing agent (0.5 μm), and the cross section of the sample was polished and etched with Kalling reagent (HF:HNO$_3$:H$_2$O = 1:6:60 vol%, t = 5–10 s). Finally, the sample was washed with alcohol solution and blow-dried. The tensile test specimen was intercepted from the sample along the deposition direction by wire cutting, and the tensile specimen was ground to 2000# using sandpaper and was then ultrasonically cleaned and prepared for use. The schematic diagram of the tensile test specimen is shown in Figure 1.

The microstructure and fracture morphology of the samples were analyzed by a ZX-10 Zeiss microscope (OM) (Zeiss, Jena, Germany) and a Hitachi S-3400N scanning electron microscope (SEM) (Hitachi, Tokyo, Japan). The microstructure of the samples was characterized by a scanning electron microscope (SEM, SU8010). The phase distribution at the interface between TiAl/V and V/TC4 was carried out by electron backscattering diffraction (EBSD). The tensile properties of different V-layer TC4/TiAl dissimilar metals deposited at room temperature were tested with a universal electronic tester (WDW-3100) with a loading rate of 0.5 mm/min.

### 3. Discussion and Conclusions

Figure 2 shows the schematic diagram and appearance of TC4/TiAl dissimilar metal samples produced by laser processing. As shown in Figure 2a, the thin-walled TC4 titanium alloy is prepared by single-pass and multi-layer methods on the TC4 titanium alloy substrate, and different layers (first, second and third layers) of metal V are deposited in the middle. Finally, the single-channel and multi-layer thin-walled TiAl alloy is deposited, and finally the TiAl-V-TC4 dissimilar alloy thin-walled sample is formed. Figure 2b–d shows the appearance and morphology of TC4/TiAl dissimilar metal samples deposited with one layer, two layers and three layers of V layers, respectively. The length × height × thickness of the molded sample is 40 mm × 45 mm × 5 mm, respectively. Through the appearance observations of the sample, the surface of the sample is formed well, and there are no macroscopic cracks or metallurgical defects.

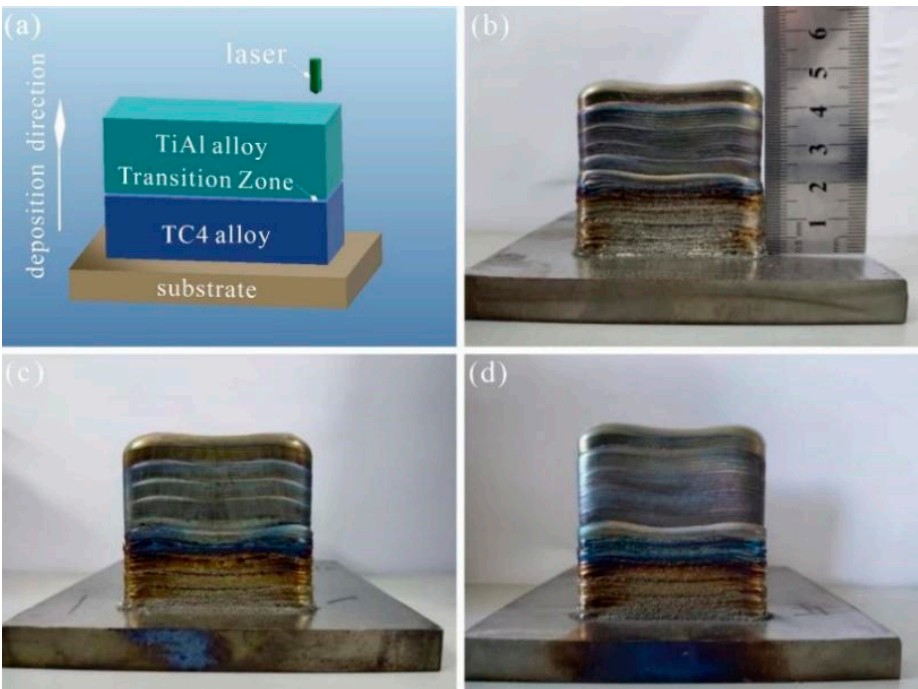

**Figure 2.** Schematic diagram and appearance of TC4/TiAl dissimilar metal samples: (**a**) Sample schematic diagram, (**b**) One-V layer, (**c**) Two-V layer, (**d**) Three-V layer.

*3.1. Effect of the Number of Deposition Layers on the Macrostructure of TC4/TiAl Dissimilar Metals*

Figure 3 shows the optical microstructure of TC4/TiAl dissimilar metals deposited with different V layers. According to the different microstructure morphologies, it can be divided into four regions, namely TiAl alloy zone, TiAl/V transition layer zone, TC4/V transition layer zone and TC4 alloy zone. It can be seen from Figure 3 that, with the increase in the number of V-layer deposits, there are no pores, metallurgical defects or microcracks in the transition layer. The organization of the TiAl alloy zone is lamellar organization, and the organization of the TC4 alloy zone is net basket-weave organization [24]. The interface structure of TC4/V and TiAl/V did not change obviously with the increase in the number of V-layer deposits [25]. The interface shape of TiAl/V is nearly zigzag, and the sedimentary transition layer combines well.

A narrow black band is seen in the TC4/V transition layer in Figure 3b,d. In order to further explore the transition region, the sample was observed under scanning electron microscope, as shown in Figure 4a,b. It can be seen from Figure 4b that the depth of the TC4/V transition zone is approximately 100 µm and the microstructure is elliptical, which is obviously different from that of the TC4 basket-weave. First of all, this is because the addition of the V element increases the undercooling of the alloy during solidification, which promotes nucleation [26]. Secondly, the solid solution of the V element produces lattice distortion in the crystal, which hinders the grain growth [27]. Due to the combined action of the above two factors, the microstructure of the alloy changes from basket-weave to ellipse. The thickness of the transition layer increases with the increase in the number of V layers, from 30 µm to 80 µm. There are no metallurgical defects and microcracks in the transition layer, which shows reliable metallurgical bonding. This shows that the V layer can effectively improve the connection problem of dissimilar metals in TC4/TiAl.

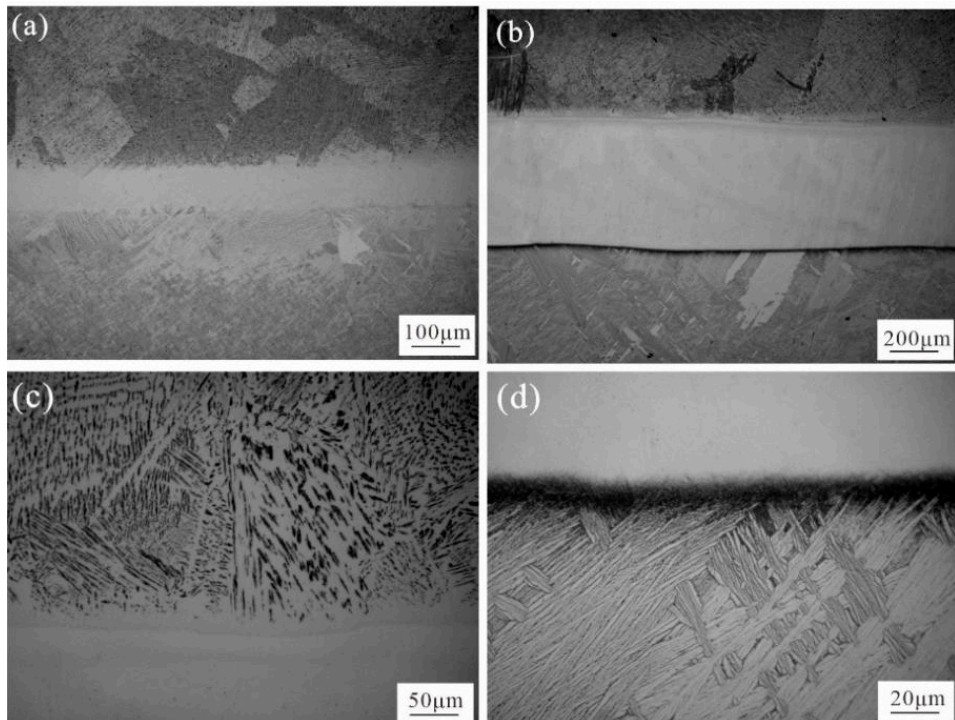

**Figure 3.** Morphology of the optical structure of TC4 and TiAl dissimilar metals: (**a**) One-V layer, (**b**) Two-V layer, (**c**) Three-layer V-layer TiAl/V domain, (**d**) Three-layer V-layer TC4/V domain.

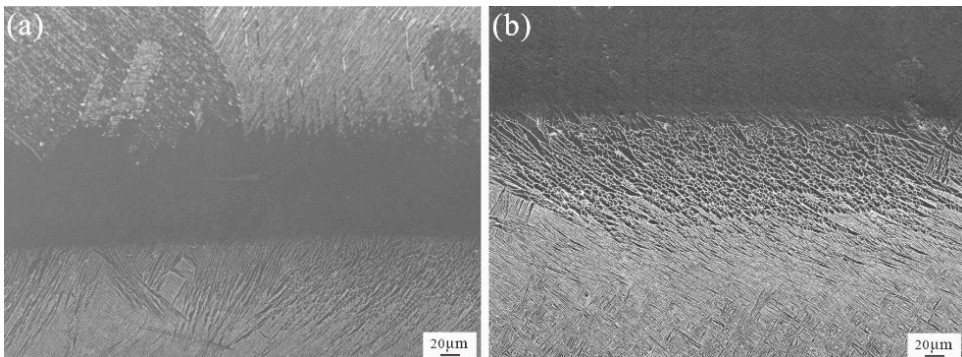

**Figure 4.** Electron microstructure and morphology of TC4/TiAl dissimilar metal: (**a**) One-V layer, (**b**) Two-layer V-layerTC4/V domain.

As shown in Figure 5 for the two-layer V-layer TC4/TiAl alloy EDS point scanning results—the upper half of Figure 5 is divided into the TiAl/V transition zone and the lower half is divided into the TC4/V transition zone—The analysis is combined with Figure 5 and Table 2,the results of the point scanning show that the content of V at point 1 is 14.18% and the content of V at point 2 is 13.90%. The higher V content at two points indicates that the solid solution diffusion of the V element in the middle layer strengthens the bonding in the transition zone. The V content at three points is as high as 51.27%, and the content of the V element in the transition zone is higher than that at far point 4, which is also as high as 49.98, indicating that the infinite solid solution effect of the V element strengthens the bonding of the transition zone in the TC4/V transition zone.

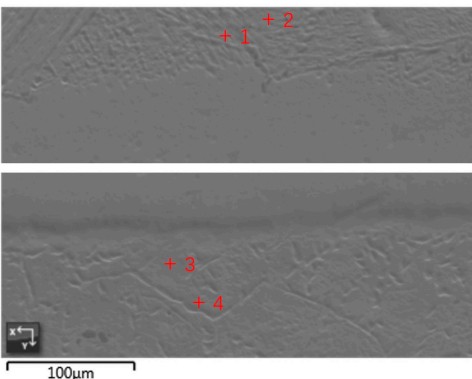

**Figure 5.** EDS analysis of the point.

**Table 2.** EDS analysis of the marking point.

| at.% | Al | Ti | V | Cr | Ni |
|------|------|------|------|------|------|
| 1 | 22.22 | 56.83 | 14.18 | 2.88 | 3.89 |
| 2 | 22.68 | 56.81 | 13.90 | 2.77 | 3.84 |
| 3 | 3.07 | 45.66 | 51.27 | 0 | 0 |
| 4 | 2.50 | 47.52 | 49.98 | 0 | 0 |

*3.2. Effect of Deposition Number on Microstructure of TC4/TiAl Dissimilar Metals*

Figure 4a shows the electron microstructure of two V-layer TC4/TiAl dissimilar metals. In order to clarify the evolution process of microstructure at the interface, the specific effects of the number of sedimentary layers on microstructural changes at the interface are explored. Combined with Figure 6a,c, it can be seen that the higher concentration of V atoms in the middle layer and the higher concentration of Ti and Al in the TC4 and TiAl regions promote the element diffusion reaction. The V element diffuses from the V layer to the TC4 alloy zone and the TiAl alloy zone, while the Ti and Al elements diffuse from both sides to the middle. Under the action of diffusion of three elements, the TC4/V transition zone and TiAl/V transition zone are formed. The contents of Al and V elements in the TC4/V transition region are higher than those in the TiAl/V transition region, which proves that there is a transition of elements at the intermediate interface. In order to further observe the phase distribution at the interface, combinations at the interface are revealed. In the EBSD results of Figure 6b, blue represents the $\gamma$ phase, red represents the $\alpha_2$ phase, yellow represents the V solid solution, green represents the $\alpha$-Ti phase and white represents the unresolved phase. The white phase is assumed to be the $\beta$ phase, because the $\beta$ phase is very fine, and the scanning step is too large during the analysis, so that the $\beta$ phase cannot be resolved [27]. The microstructure at the interface is mainly composed of the $\gamma$, $\alpha_2$, V, $\alpha$-Ti and $\beta$ phases. The microstructure of the TC4/V interface is mainly composed of $\alpha$-Ti, $\beta$ phase and V solid solution. In the TC4 alloy zone, because the V element is the stable element of the $\beta$ phase in the dual-phase titanium alloy in the TC4 alloy, and the V element is infinitely dissolved in the $\beta$ phase, a connection without metallurgical defects is formed in the TC4/V transition zone [27,28]. In the TiAl/V transition region, the addition of the V element does not change the phase composition of the transition region [29,30]. The good metallurgical bonding is due to the fact that the V phase infiltrates the grain boundaries of the $\gamma$ and $\alpha_2$ phases, which makes the V elements combine closely with the $\gamma$ and $\alpha_2$ phases in the form of a solid solution [31].

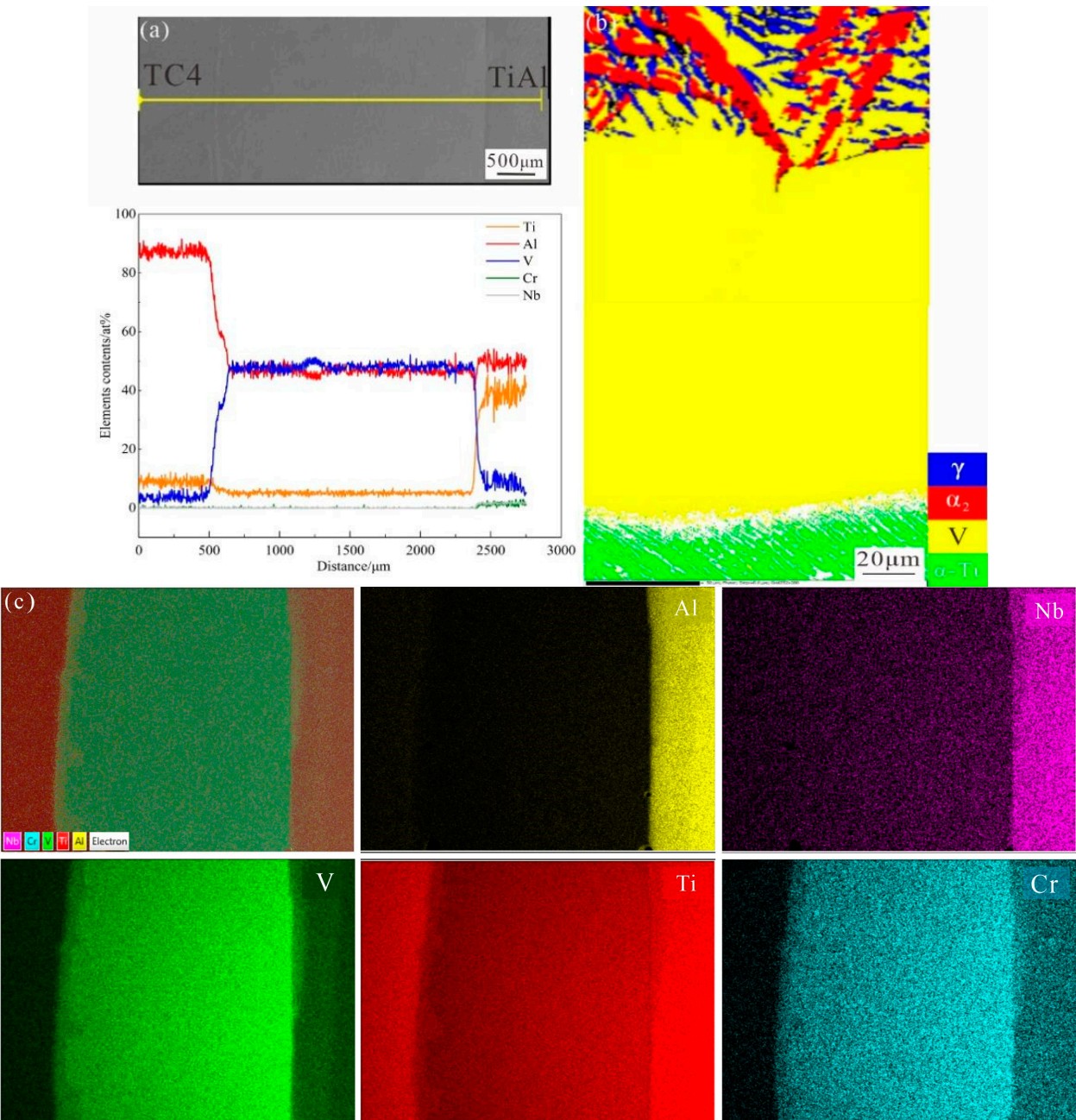

**Figure 6.** Characterization of cross-sectional microstructure of dissimilar metal joints in TC4/TiAl: (**a**) EDS line scan map of TC4-V-TiAl junction; (**b**) Phase distribution diagram at TC4-V-TiAl connection; (**c**) EDS scan atlas of TC4-V-TiAl junctions.

### 3.3. Effect of Deposition Number on Micromechanical Properties of TC4/TiAl Dissimilar Metals

Figure 7 shows the ultimate tensile strength chart. Figure 8 shows the stress–strain and fracture morphology of TC4/TiAl dissimilar metals with one layer, two layers and three layers. It can be seen from the stress–strain diagrams of Figures 7 and 8a–c that there is no obvious yield stage before fracture. The average tensile strength and elongation of deposited one-layer, two-layer and three-layer TC4/TiAl dissimilar metals at room temperature are 405 MPa, 483 MPa, 350 MPa and 0.24%, 0.35%, 0.16%, respectively. In the comparison of the above data, it can be seen that, with the increase in the number of deposition layers, the tensile properties of the dissimilar alloy at room temperature first increases and then decreases. The comprehensive mechanical properties of the two-layer

V-layer TC4/TiAl dissimilar alloy are the best, and the tensile strength and elongation are increased by 19.2% and 45%, respectively, compared with the first V-layer. It can be seen from Figure 8d–f that there are river patterns, tearing edges and cleavage surfaces on the fracture surface, and the fracture mechanism is a quasi-cleavage fracture. With the increase in the number of deposited V layers, the proportion of cleavage planes first decreased and then increased, while the proportion of tearing edges first increased and then decreased. There are two main reasons for this. One is that the addition of the V element can produce the effect of grain refinement, the other is that the solid solution of the V element leads to an increase in the number of grain boundaries and appropriate lattice distortion, which hinders the movement of dislocations in the alloy in the process of tensile deformation, which increases the difficulty of deformation, thus improving the tensile strength of the alloy [32]. However, the addition of too many V elements will seriously weaken the mechanical properties of the sample, which can be seen in Figure 8f. In Figure 8f, it can be seen that the proportion of cleavage surfaces is the largest and the river pattern is shallow, indicating an increase in brittleness of the alloy. The increase in brittleness is determined by many factors. Firstly, the three V layers lead to too much V element, and the solid solution of the V element increases the effect of lattice distortion, which leads to greater stress concentration. Secondly, the segregation of the micro-composition of the metal solution in the non-equilibrium solidification leads to the non-uniform composition of the alloy [33,34]. The fundamental change in properties between the two- and three-layer samples is due to the V-layer being too thick, which weakens the solid solution strength and causes the tensile samples to fracture in the vanadium layer.

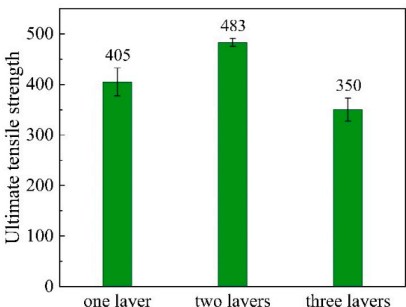

**Figure 7.** Ultimate tensile strength chart.

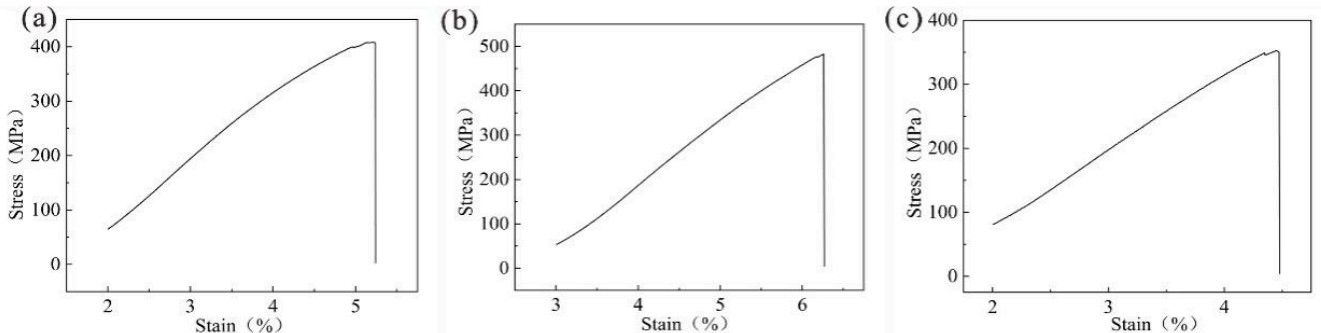

**Figure 8.** *Cont.*

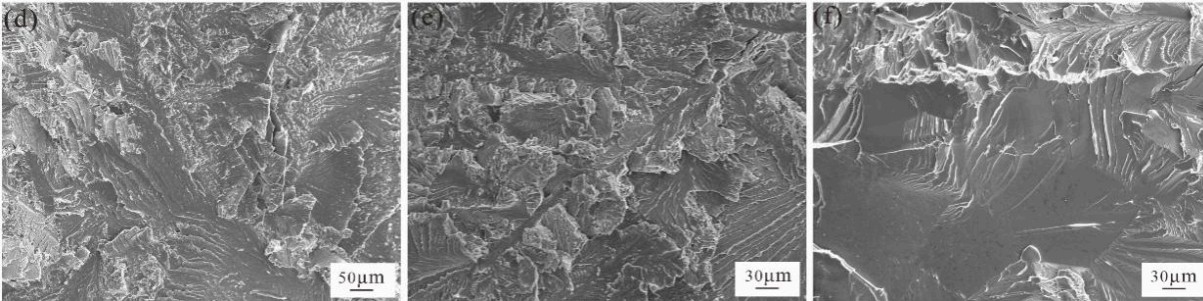

**Figure 8.** Stress–strain curve and fracture diagram: (**a**–**c**) One-layer, two-layer and three-layer stress-strain diagrams; (**d**–**f**) Fracture topography of one layer, two layer and three layers.

Finally, this leads to a decrease in tensile strength and plasticity. The reason for the fracture mechanism is the transition from quasi-cleavage fracture to brittle fracture.

**4. Conclusions**

A sample of TC4-V-TiAl alloy without metallurgical defects was successfully prepared, and the interface between TiAl/V and TC4/V was well connected without microcracks. The microstructure of the TC4/V interface was mainly composed of β-Ti and V solid solution. The interface microstructure of the TiAl/V was mainly composed of the γ and α$_2$ phases and V solid solution.

The mechanical properties of TC4/TiAl dissimilar alloys bonded with one, two and three V layers were compared. The comprehensive mechanical properties of two-layer V-layer connected TC4/TiAl dissimilar alloy samples are the best, and their tensile strength and elongation are 483 MPa and 0.35%, respectively. The fracture mechanism is a quasi-cleavage fracture. Compared with a V-layer TC4/TiAl dissimilar alloy, the tensile strength and elongation are increased by 19.2% and 45%, respectively. The mechanical properties of the three-layer vanadium layer (483 MPa, 0.35%) are the worst. The reason for the fracture mechanism is the transition from quasi-cleavage fracture to brittle fracture.

**Author Contributions:** Conceptualization, H.W. and Z.L.; methodology, H.W. and Z.L.; software, J.L.; validation, G.Y.; formal analysis, W.W. and G.Y.; investigation, H.W.; resources, Z.L.; data curation, Z.L.; writing—original draft preparation, H.W. and Z.L.; writing—review and editing, H.W. and Z.L.; visualization, J.L.; supervision, Z.L.; funding acquisition, H.W., W.W. and Z.L. All authors have read and agreed to the published version of the manuscript.

**Funding:** This research was funded by the Doctoral Start-up Foundation of Liaoning Province (grant number: 2023-BS-195), and the Basic Scientific Research Project of Liaoning Provincial Department of Education (grant number: LJKMZ20220960).

**Institutional Review Board Statement:** Not applicable.

**Informed Consent Statement:** Not applicable.

**Data Availability Statement:** Data are unavailable in this study.

**Conflicts of Interest:** The authors declare no conflict of interest.

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
