# Peer review of "Effect of Vanadium Layer on Microstructure and Properties of TC4 (Ti-6Al-4V)/TiAl (Ti-48Al-2Cr-2Nb) Dissimilar Metals Produced by Laser Additive Manufacturing"

_coatings, doi:10.3390/coatings13091638_

Round 1

Reviewer 1 Report

I found the article interesting and well-written. The literature is up to date, the methods used are thoroughly explained and the results are clearly represented and discussed. The article is not completely original, as similar research has already been conducted, but it contributes to the field of research.

Some minor revisions are necessary. In paragraph 3.2. sentences "In order to clarify the evolution process of microstructure at the interface." and "In order to further observe the phase distribution at the interface." are unfinished. You need to complete them, i.e. make a point at the end or connect them with a previous sentence, because this way it looks like a sentence fragment rather than a complete sentence. Additionally, the last sentence in the discussion, as well as the last sentence in the conclusion were a bit confusing to me: "The reason of the fracture machine is the transition from quasi-cleavage fracture to brittle fracture." Instead of machine, do you mean mechanism?

Author Response

Thank you for your valuable comments. Regarding your comments, I will reply you one by one in the attachment. Thank you very much again

Reviewer 2 Report

coatings-2560419

The work deals with the joining of two dissimilar titanium alloy materials by means of a laser additive manufacturing process. The authors have addressed a topic of interest and the work seems to be well prepared. 

However, I believe that the authors should address some aspects to improve it. 

Title

The authors should include when talking about TC4 and TiAl, at least in brackets, the composition of the alloys, I suggest that they include the following information in the title: TC4 (Ti-6Al-4V) and TiAl (Ti-48Al-2Cr-2Nb).

Introduction

The authors should explain in more detail and at greater length what are the possible applications of TC4/TiAl bonding. It is not clear what is expected from this bonding in specific cases. 

There is a need to include more extensive information on other techniques that could be used to join these dissimilar materials. Expand the state of the art in the introduction. 

On the other hand, the authors should justify more conclusively that this additive manufacturing technology is more suitable than other additive manufacturing technologies. 

Finally, it is worth justifying the choice of the bonding element for these materials, i.e. vanadium. It would also make sense to know the criteria for the selection of layer thicknesses and the number of layers. 

Methodology

The authors indicate that the TC4 surface was prepared with a grinder and then cleaned. There is no surface information on the condition of that surface, was the roughness not measured, and do the authors not think that it could be a highly relevant factor in securing the bond? Justify your answer

How have the parameters for the laser treatment been selected?

What type of specimens and how were these specimens manufactured for the universal tensile test? It would be interesting to provide photographs of the specimens. 

Results

The numerical data assigned to the mechanical properties results lack dispersion and should be included.

Author Response

(The authors gave the same response as above.)

Reviewer 3 Report

This manuscript presents an experimental work on the additive manufacturing of dissimilar materials. Although it is an interesting topic, several parts of this work are incomplete and some parts of the analysis are also insufficient. Thus, it cannot be recommended for publication in its current form. The main issues with this work are:

The AM method should be described in more details. The characterization "Laser additive manufacturing" is far too general and may imply various different methods.

The Introduction section should include more works relevant to the topic of the paper and the authors should avoid mentioning multiple works without discussing them.

The authors did not justify why they chose vanadium as the material for the interface.

The authors did not mention the standard used for the mechanical testing, the type of specimen and how it was cut from the printed parts.

One significant drawback of this work is that actually no investigation was conducted for a single process parameter. The authors conducted only a single experiment with three different numbers of layers, without varying the process conditions or without determining the optimum settings. 

In Section 3, the authors state that the observation of the produced part indicates no macroscopic cracks or other defects. However, in Figure 1, it can be observed that the quality of the produced parts is not optimal and it cannot be clear that it lacks any defects.

In line 144, the authors state "the microstructure of the alloy changes from basket to ellipse". The terminology should be further checked, e.g. "basket-weave" instead of "basket".

The authors did not provide any specific measurements regarding the microstructure, e.g. heat affected zones or grain size. 

The discussion about the mechanical properties is insufficient. The authors did not include any error bars resulting from multiple tests.  The authors did not provide a definite reason for the radical variation of properties between the specimen with 2 layers and 3 layers. More tests are required in order to provide a clear critical value for the optimum mechanical properties, probably by varying also the layer height of the deposited material and not only the number of layers with a fixed height. 

Moderate revisions are required.

Author Response

(The authors gave the same response as above.)

Round 2

Reviewer 2 Report

The authors have provided adequate answers to the issues raised. Thank you very much.

For my part the document can be published. 

Author Response

Dear reviewer,

Hello!

Thank you very much for your comments on our articles. These valuable comments are of great help to the revision and improvement of our manuscript.Thank you for your valuable comments, thank you very much again.

Reviewer 3 Report

The authors performed only a few minor modifications to their manuscript, but still they have not answered adequately several questions. Thus, this work cannot be recommended for publication in its current form. Some of the main issues of this work are:

The authors have not still mentioned directly the laser based AM process which was used e.g. SLM. 

More works should be mentioned and discussed in the Introduction section.

The authors still did not mention the standard used for the experiments.

The number of experiments and the discussion is still not sufficient.

Results from detailed microstructured measurements are required instead of mentioning the range of grain size.

Moderate revisions are required.

Author Response

Thank you for your valuable comments, thank you very much again.

Round 3

Reviewer 3 Report

The authors performed some of the proposed changes to their manuscript. It can be now considered for publication. Before submitting the final mansucript, they should mention which standard was used for the tensile tests e.g. based on  ISO or ASTM.

Minor changes are required.